# Is Task-Agnostic Explainable AI a Myth?

**Alicja Chaszczewicz** [1]

## Abstract

Our work serves as a framework for unifying the challenges of contemporary explainable AI (XAI). We demonstrate that while XAI methods provide supplementary and potentially useful output for machine learning models, researchers and decision-makers should be mindful of their conceptual and technical limitations, which frequently result in these methods themselves becoming black boxes. We examine three XAI research avenues spanning image, textual, and graph data, covering saliency, attention, and graph-type explainers. Despite the varying contexts and timeframes of the mentioned cases, the same persistent roadblocks emerge, highlighting the need for a conceptual breakthrough in the field to address the challenge of compatibility between XAI methods and application tasks.

## 1. Introduction

The development of explainability methods, or their incorporation into AI models, has often been motivated by the aim of enhancing user trust in the model. Users may view explanations similarly; for instance, a survey of clinicians by (Tonekaboni et al., 2019) revealed that they perceive explainability as *"a means to justify their clinical decision-making."*

Despite numerous results highlighting limitations of explanations (Adebayo et al., 2018; Rudin, 2019; Slack et al., 2020; Tomsett et al., 2020; Neely et al., 2021; Bilodeau et al., 2022), they continue to be considered in high-risk decision contexts, such as healthcare (Loh et al., 2022).

While various methods have been developed and used, we observe a missing link between the theoretical or technical problems that these methods address and the way in which practitioners may hope to apply them – to assist with actual real-world tasks and decision-making. Our work elucidates why XAI methods are not, in general, reliable decision justification enablers.

We focus on a broad class of explanations, for which a complex uninterpretable machine learning model is first trained, and then XAI aims to "explain" its predictions for a given input. We expose the unstable foundations of the examined XAI methods by presenting three illustrative research case studies that disclose a prevailing pattern across various machine learning tasks, data domains, and time frames. This pattern can be succinctly described as follows:

- Stage 1: XAI method is developed, but its utility is presented in a simplistic setup; theoretical guarantees relevant to potential real-world tasks are missing.

- Stage 2: As variations of the method are developed, an alternative perspective allows for a more comprehensive evaluation, revealing the method's limitations and bringing its reliability into question.

- Stage 3: The inconclusive, and sometimes contradictory results, provided by subsequent evaluation techniques, metrics, and theoretical considerations, make it challenging to determine the best-suited method variation for specific practical tasks and to identify when the method is truly effective and dependable. It demonstrates that the methods cannot be considered reliable task-agnostic explainers.

Our work is not a survey but can serve as a framework for unifying the challenges in the development process of contemporary XAI, which tends to follow the listed above stages. We hope this framework assists researchers, current practitioners, and potential XAI users in a better contextual understanding of what XAI currently is and is not.

In the following, we investigate three XAI research trajectories covering image, text, and graph data, which utilize saliency, attention, and graph-based explanation methods, respectively. Each research story is presented in a distinct section, wherein the subsections directly correspond to the stages delineated in the framework we propose.

[1]Department of Computer Science, Stanford University. Correspondence to: Alicja Chaszczewicz <alicja@cs.stanford.edu>.

*Workshop on Interpretable ML in Healthcare at International Conference on Machine Learning (ICML)*, Honolulu, Hawaii, USA. 2023. Copyright 2023 by the author(s).

## 2. Saliency explainers & image data

The first research story is a classic of explainable AI literature. Natural to modern deep learning pipelines and straightforward to implement, saliency methods have quickly become a popular XAI tool (Samek et al., 2021). The idea behind them is to apply the first-order approximation of the amount of influence that the input data has on the machine learning model output.

The saliency methods can be used across different data domains. Originally, they were introduced for computer vision tasks and are still frequently applied in this context. In the following discussion, we also focus on image data.

### 2.1. Stage 1: Idea and methods

The vanilla saliency method (Simonyan et al., 2014) boils down to a gradient computation of output with respect to input data. The result is a map with the same dimensionality as the input, showing each input part's detailed "importance" (here, gradient). Taking images as an example, each pixel gets its attribution value, which is a gradient of output with respect to this pixel. The end product is a saliency map (Figure 1).

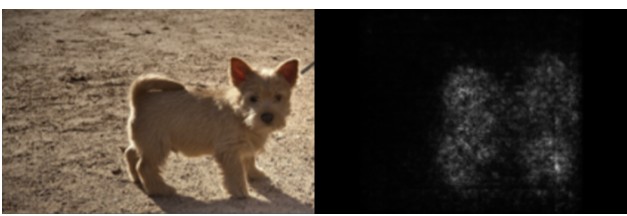

*Figure 1.* A Convolutional Neural Network was trained to classify images. The dog image (left) is fed into the model. The resulting saliency map (right) for the predicted class is obtained by a gradient computation through a single backward pass (right). The intensity of each pixel represents the magnitude of the output gradient with respect to this pixel. Figure adapted from (Simonyan et al., 2014)

The initial gradient idea sparked a fruitful investigation that aimed to make the gradient-based influence approximation better and the resulting maps "clearer" (Figure 2). The engineered methods included combining gradients with input data values, smoothing gradients by adding noise and averaging, integrating gradients over a path from a baseline image (e.g., a black image), zeroing out negative gradients while backpropagating or combining gradients and final convolutional layer feature maps (GradientxInput (Shrikumar et al., 2017), SmoothGrad (Smilkov et al., 2017), Integrated (Sundararajan et al., 2017), Guided Backpropagation (Springenberg et al., 2015), GradCAM (Selvaraju et al., 2016))

None of those methods was evaluated in a quantitative way, as pointed out by the paper introducing SmoothGrad, the

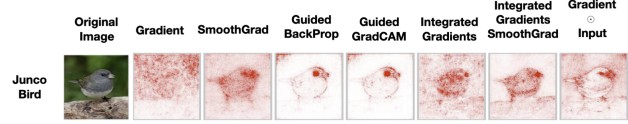

*Figure 2.* Saliency maps for an input image of a bird (left) generated by different saliency methods. While the vanilla gradient method output is noisy, the other methods "improve" the map visually. Figure adapted from (Adebayo et al., 2018)

latest of the listed above methods: *"Since there is no ground truth to allow for quantitative evaluation of sensitivity maps, we follow prior work and focus on two aspects of qualitative evaluation."*(Smilkov et al., 2017) The two mentioned aspects relied on purely visual qualitative analysis.

At that stage, evaluations and comparisons prioritized visual aspects. Although many methods drew inspiration from theoretical concepts, they lacked a direct connection and guarantees related to how well the "explanations" correspond to the model's "reasoning" process.

### 2.2. Stage 2: Experiments questioning reliability

Throughout that development phase, assessments were carried out to determine if the saliency maps generated aligned with the method's initial goal to "explain" the model. The method's reliability was called into question through meticulously designed experiments.

A fresh perspective on the evaluation and validity of the saliency methods came in the work *"Sanity Checks for Saliency Maps"* (Adebayo et al., 2018). The paper introduced two sanity checks that could reveal if a given saliency method is not working as intended, i.e., serving as an explanation. The two checks analyzed how the saliency maps change

- when the model weights are randomized;

- when the model is trained on data with random labels.

The results of the novel quantitative empirical analysis were alarming. Only two of the tested methods fully passed the checks. The other ones produced visually very similar results **independent** of whether the model was properly trained, had randomized weights or was trained on **random** labels. For example, if saliency maps for the classification of a digit 0 were generated for a model trained on the MNIST dataset with random labels, the "explanations" looked as in Figure 3. Many saliency methods produced sensible "explanations", while the network was unable to classify with a better than random guessing accuracy.

One might point out that this analysis is still purely vi-

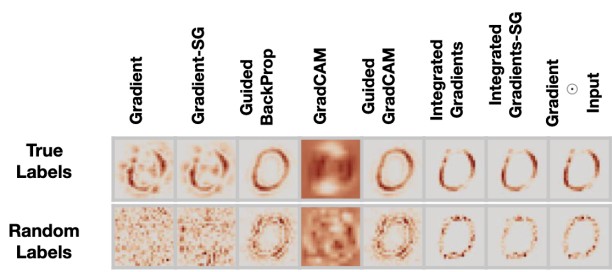

*Figure 3.* Saliency maps produced for a MNIST image of digit zero. For the majority of the methods, the explanations produced for a properly trained model (top row) look very similar to maps generated for a model trained on random labels (bottom row). Figure adapted from (Adebayo et al., 2018)

sual. Indeed, the authors also compared pre- and post-randomization saliency maps quantitatively, and, for some of the methods, the metrics were telling a different story than the visuals, i.e., in some cases, some metrics distinguished between the two scenarios (pre- and post-randomization) while visually the produced maps seemed very similar.

## 2.3. Stage 3: Beyond original visual assessment

It became evident that solely relying on a visual comparison of methods is insufficient for assessing their effectiveness. As a result, new metrics and tests have been developed to evaluate the connection between methods and their corresponding tasks. However, designing these metrics presented challenges.

### 2.3.1. METRICS AND GROUND TRUTH

A natural next step might be to find "good metrics", based on which it could be determined whether or not a given saliency method is best at producing "relevant explanations". This, however, is a tough problem when ground truth explanations are unknown; many suggested metrics for saliency map evaluation have been shown to be unreliable (Tomsett et al., 2020; Hedström et al., 2023). Another approach is to use a dataset with known ground truth. Under such a scenario, we can see which method recovers the ground truth best. The results, however, do not necessarily generalize to a real-world dataset of interest (Yang & Kim, 2019). Moreover, a "golden" metric would not be a complete problem solution. A visual inspection might be the actual way how users interact with an explanation method. If this is the case, we would need to ensure that conclusions based on visual perception are helpful for a task of interest; a proper metric would be insufficient.

### 2.3.2. SANITY CHECKS

Not only metrics but also the tests themselves are so far unable to tell the whole story. A sanity check or an evaluation protocol is not a task-independent indicator of the saliency method's validity. We should take into account how the saliency method, model, and task interact with each other, as this might confound the final validity or invalidity conclusion. For example, even if the network weight randomization sanity check fails, it actually does not determine that the tested saliency method is generally insensitive to model weights. It rather shows that the inductive bias in the network is strong enough that for some specific combinations of saliency method and task, the method produces "explanation-like" looking outputs (Yona & Greenfeld, 2021). Moreover, the above sanity checks are not exhaustive, and other tests have been proposed (Kindermans et al., 2019).

Saliency maps are explanations that can be generated for any differentiable models. However, in light of these limitations, providing concrete guarantees or applying rigorous assessment tools to accurately establish their comparative effectiveness and reliability is currently unattainable.

## 3. Attention explainers & textual data

Let us consider another explainability method based on potentially "interpretable" attention components of a model.

The attention mechanism (Bahdanau et al., 2015; Vaswani et al., 2017) is a technique used in neural networks to focus on certain parts of the input data selectively. By emphasizing or ignoring certain information, attention can improve model performance and, possibly, model explainability potential. While attention "explanations" are currently used across different tasks, attention was initially introduced as part of natural language processing models. In the following, we focus on the NLP domain.

### 3.1. Stage 1: Idea and methods

The attention weights that determine focus on the given input part can be considered "importance" weights, i.e., the bigger the weight, the more critical the input element is (Figure 4)[1]. Highlighting input elements with the greatest attention scores has been used in numerous research papers to visualize neural network "reasoning process" and to somehow "explain" it, for example (Xu et al., 2015; Choi et al., 2016; Lee et al., 2017; Mullenbach et al., 2018; Chen

---

[1]The attention mechanism is used in different layers, and the input to the attention module is not necessarily the model input. In the following, we will focus on attention weights with respect to NLP model inputs. The attention weights "interpretation" occurs in a much broader context (e.g., various data modalities, different attention module types, variability across layers)

et al., 2020).

**Question**: Where is Sandra ?
**Original Attention**:John travelled to the garden . Sandra travelled to the garden

*Figure 4.* A model with the attention module was trained to solve a question answering problem. For the presented question, the greatest attention weight was attributed to the word *garden*. Assuming attention correlates with importance, this might indicate that the word *garden* was crucial for the model prediction. Example adapted from (Jain & Wallace, 2019)

At that stage, interpreting weights in the attention module as "importance" is abstract and ill-defined. Used to highlight key input components, this high-level concept aids users in identifying crucial input aspects but falls short of providing a thorough understanding of underlying mechanisms and the model's "reasoning".

## 3.2. Stage 2: Investigations challenging dependability

If the attention weights are employed as explanations for the model's "reasoning process", it is essential to assess them as such and examine their correlation with the model's actual behavior.

The work *"Attention is not explanation"* (Jain & Wallace, 2019) raised concern about the ongoing natural adoption of the attention mechanism for explainability purposes. The authors posed the following questions:

- Do attention weights correlate with other feature importance measures?

- Do alternative attention weights significantly change model predictions?

Based on the performed empirical tests, the answer to both was *"No, largely they do **not**."*

To answer the first question, the work compared computed attention scores with importance measures based on gradients (see Section 2) and leave-one-out methods – one input element was removed, and the amount of change in the output was attributed as the importance of this element. The experiments demonstrated no consistent correlation between the attention scores and those two other importance attribution methods.

The second question was addressed by assessing the output change in the case of permuted attention weights and adversarial attention weights, which were found by maximizing the distance from the original attention scores while keeping the model output similar. The two tests showed there exist many alternative attention weights configurations (found

by shuffling or adversarially) that produce similar model outputs (Figure 5).

**Question**: Where is Sandra ?
**Original Attention**:John travelled to the garden . Sandra travelled to the garden
**Adversarial Attention**:John travelled to the garden . Sandra travelled to the garden

*Figure 5.* The original attention mechanism focuses on the word garden, which refers to the place to which Sandra traveled. After adversarially shifting the focus to the word irrelevant to the question asked (the place where John traveled), the model output remains almost the same. Example adapted from (Jain & Wallace, 2019)

## 3.3. Stage 3: Inconsistencies in evaluation methods

Previously established evaluation methods were not comprehensive; further research and findings demonstrate numerous nuances related to the evaluation process and the evaluation tasks. Various papers have attempted to make attention mechanisms "more explanatory", while others suggest that attention may serve as an explanation for some tasks but not for others.

### 3.3.1. ADVERSARIAL WEIGHTS

The mentioned above conclusions were shortly questioned by *"Attention is **not not** explanation"* (Wiegreffe & Pinter, 2019). The authors found the permuted and adversarial attention weights experiments insufficient to support the claim that attention is not an explanation.

Therefore they suggested new diagnostic tests. The experiments were performed on the datasets filtered by a procedure that checked if a model with an attention module was significantly better than a model with fixed uniform weights. If the test outcome was negative, the authors did not consider the given dataset further, claiming that *"attention is not explanation if you don't need it."*.

Then an adversary network was trained for the whole dataset, unlike (Jain & Wallace, 2019), who found adversarial weights per individual data instance. This change in adversary computation was justified by the statement that manipulating part of the trained model per instance did not actually demonstrate a full adversarial model able to produce claimed adversarial explanations.

The new experimental analysis showed that the fully adversarial model did manage to find adversarial attention weights, but they were not as distant from the original weights as in (Jain & Wallace, 2019). Moreover, the adversarial weights seemed to have less encoded information when tested with the introduced diagnostic tool, which compared the performance of the MLP network with the final layer averaging token representations using original atten-

tion vs. adversarial weights. The traditional original attention weights resulted in better MLP evaluation scores. The authors conclude that in the current form, attention weights cannot be treated as *"one true, faithful interpretation of the link"* between model inputs and outputs but that the experimental analysis done so far does not show that attention is not an explanation.

### 3.3.2. CORRELATION WITH OTHER ATTRIBUTIONS

While (Wiegreffe & Pinter, 2019) found the experimental analysis for agreement between attention importance and other feature importance measures valid, the later work (Neely et al., 2021) claims that measuring agreement is not a proper evaluation method. Consistency as evaluation implicitly assumes that one of the methods is nearly ideal, since we aim to find a high correlation with it. This is not necessarily true. (Neely et al., 2021) show little correlation between a range of feature attribution methods.

### 3.3.3. FURTHER POINTS

(Serrano & Smith, 2019) claim attention weights do not form good explanations as it is easier to make a model flip its decision by removing features with higher gradient importance rather than ones with high attention weights. Conversely, (Vashishth et al., 2019) point out that cited above (Jain & Wallace, 2019; Wiegreffe & Pinter, 2019; Serrano & Smith, 2019) all evaluated attention explainability for a single NLP task type. (Vashishth et al., 2019) claim that in other NLP tasks, attention is "more explainable". Some researchers attempt to modify attention mechanisms so that it better reflects the model's decision process (Mohankumar et al., 2020).

The debate, on whether attention is a good explanation and, if so, when, is still ongoing (Bibal et al., 2022). The discrepancies observed among evaluations result in no guarantees when applying this explanation method in task-agnostic settings, given the numerous variations in behaviors encountered.

## 4. Graph-type explainers & graph data

Let us continue to the most recent research story, which in contrast to saliency 2 and attention 3 starts with some set of clearly defined quantitative "explainability" evaluation tests.

As Graph Neural Networks (GNNs) emerged as a new paradigm of machine learning on graphs, a need for XAI methods for GNNs arose.

The saliency methods (Section 2) can be applied to GNNs, as pioneered by (Pope et al., 2019; Baldassarre & Azizpour, 2019). Attention weights XAI ideas (Section 3) can be used

as well and transfer directly to the GNN domain when Graph Attention Networks are the model of choice (Velickovic et al., 2018).

Although these two categories of explainability techniques can be employed, the inherent nature of graphs that combine feature information and topological structure gave rise to many XAI ideas targeting graph-based data.

### 4.1. Stage 1: Idea and methods

The first method providing explanations specifically for GNNs was GNNExplainer (Ying et al., 2019). GNNExplainer aims to find a compact subgraph that is the most influential driver behind a GNN prediction. This subgraph is determined by the edge and feature masks found in a gradient-based optimization procedure.

To evaluate the method (Ying et al., 2019) suggest a set of synthetic datasets. The graphs in the datasets are generated randomly, and the ground truth classes are dictated by predefined motifs planted in the graphs (Figure 6). The explainers can then be evaluated based on their ability to find those motifs. (Ying et al., 2019) evaluate predictions also qualitatively on two real-world datasets.

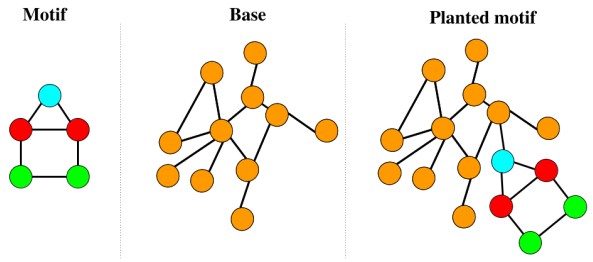

*Figure 6.* Generation of a synthetic graph XAI evaluation dataset introduced by (Ying et al., 2019). First, a special motif is defined, here, a house shape (left). Then a random base graph is created (middle). Finally, a motif is planted in the generated random graph (right). Node classes are defined based on their different structural roles (colors) in the graph, optionally also on node features.

The subsequent work, such as PGExplainer (Luo et al., 2020), PGMExplainer (Yuan et al., 2021), and SubgraphX (Vu & Thai, 2020), followed (Ying et al., 2019) and evaluated GNN explainers on the suggested synthetic datasets or their close variants. The set of real-world datasets employed displayed variation. One dataset frequently but not always utilized (Dai et al., 2022) to evaluate GNN explainers is MUTAG (Debnath et al., 1991) (Figure 7). Real-world datasets usually do not have ground truth "explanations", therefore the evaluation is either qualitative or based on some metric showing how the model's predictions change after manipulating the original graph, e.g., by removing the explanation

part from the graph or taking the produced explanation as input, without the rest of the graph.

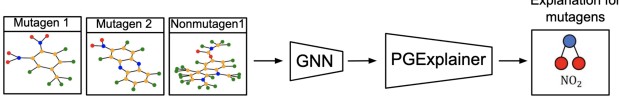

*Figure 7.* The MUTAG dataset (Debnath et al., 1991) consists of graphs representing different molecules. Some of those molecules are mutagenic. The machine learning task is to classify the graphs as mutagenic or not. To solve this problem, a GNN can be trained. An explanation module (here PGExplainer) is expected to detect subgraphs whose chemical properties determine mutagenicity. Figure from (Luo et al., 2020)

At this phase, the construction of graph explainable artificial intelligence (XAI) involves artificial tasks and an absence of definitive assurances.

### 4.2. Stage 2: Navigating datasets limitations

(Faber et al., 2021) find the state of evaluation of former GNN explainers' work not satisfactory. They find evaluation procedures with synthetic datasets and also with real-world datasets limited and unreliable. They argue that former evaluations on real-world datasets done by manipulating the input graphs (for example, removing the explanation subgraph and checking how significantly the model output changes) are improper. For instance, in the case of more complex explanations, if a class corresponds to the presence of motif A or motif B, when both motifs are in a graph, removing either of the motifs would not change the model output. This would indicate that the removed motif is not important for the prediction, while it actually is. Moreover, they point out that, similarly to other data domains, manipulating input graphs changes input data distribution, which might affect the evaluation procedure.

Therefore they recommend using synthetic benchmark datasets with known ground truth. However, they find existing synthetic datasets and evaluation procedures prone to several pitfalls that prevent a fair explainers comparison:

- **Bias term** GNN model can learn a bias term; for example, by default, it could output class 0, which corresponds to some planted motif A, and output class 1 if found evidence of the existence of motif B. Therefore, explainers will have no success finding motif A as an explanation subgraph because of the bias terms; this motif A is simply not used by the GNN reasoning process.

- **Redundant evidence** If detecting only part of the motif suffices to produce a correct output, we should not

expect an explainer to be able to find a full explanation subgraph.

- **Trivial explanation** The existence of a trivial explanation, such as nearest neighbors nodes or personalized Page Rank (Page et al., 1999) attributions, might result in a positive evaluation of a GNN explainer, while it might be doing something trivial not connected to the GNN "reasoning".

- **GNN performance and alignment** In case GNN performs poorly on a given task, the fact that an explainer does not find a proper explanation may be due to imperfect GNN training and not an underperforming GNN explainer. Moreover, if architecture enforces processing of focus on the non-explanation graph part, a GNN explainer will be unable to uncover the pure explanatory part.

Three synthetic benchmark datasets introduced by (Faber et al., 2021) aim to avoid the above pitfalls. The authors evaluate several saliency gradient-based methods (Section 2) and two graphs targeting explainers GNNExplainer and PGMExplaner. The latter did not perform well on the proposed benchmarking datasets, while the former gave "mixed results", with no method mastering all three datasets.

### 4.3. Stage 3: Lack of evaluation standardization

As the field of GNN explainers grew, several experimental research papers attempted to evaluate a set of developed GNN explainability methods.

(Agarwal et al., 2023) develop a flexible synthetic data generator taking (Faber et al., 2021) recommendations into account. The best performer among tested methods for the generated dataset is graph-data-focused SubgraphX. However, gradient-based methods take the lead when the experiments are conducted for three real-world datasets (modified for unique explanations).

(Amara et al., 2022) agree with (Faber et al., 2021) and argue that the initial set of synthetic datasets *"gives only poor informative insight."* They show that simple baselines (e.g., Page Rank) almost always beat the developed, more sophisticated explainers on those datasets. Moreover, they evaluate GNN explainers on ten real-world datasets without ground-truth explanations. In contrast to (Faber et al., 2021), they perform an experimental evaluation using metrics quantifying GNN model output change when the explanatory part is removed or the only part left. Overall they find that the simple gradient saliency method performs best. Additionally, they present a decision tree of what explainability method is recommended depending on the metric of interest and explanation type. However, when testing the framework on a new dataset, the best-performing explainer turned out to

be different from all the methods in the suggested decision tree for any choice of metrics and any explanation type.

When testing on a different set of real-world datasets without ground truth (Yuan et al., 2022) conclude that graph-data focused SubgraphX method *"outperforms the other methods significantly and consistently."* On the contrary, a study of the use case of cyber malware analysis finds that the gradient-based methods perform best (Warmsley et al., 2022).

(Longa et al., 2022) introduce a set of synthetic datasets with known ground truth, aiming to follow (Faber et al., 2021) guidelines. The experimental analysis shows that which method performs best is not constant and changes between datasets. Not only datasets but also a GNN architecture influences strongly performance of explainability methods, which work well for some architectures and fail for others.

All of the above experimental studies use various evaluation protocols, a changing set of explainability methods, a different set of synthetic or real-world datasets, and various GNN model architectures.

The above, therefore, indicates that there are no comprehensive guidelines available for practitioners and that the field has yet to reach a consensus on evaluation methods for graph XAI.

## 5. Summary

### 5.1. Lack of reliability guarantees for XAI methods

We established a unifying framework to address XAI challenges, identifying a recurring three-stage pattern in method development, regardless of data type, explanation unit, or timeline. Initially, XAI methods lack real-world applicability due to simplistic presentation and missing guarantees. Assessing emerging variations reveals limitations and reliability concerns. Inconsistent evaluation results challenge the determination of tasks for specific methods, dismissing universally applicable XAI methods.

We illustrated the pattern in three cases – image, textual, and graph data with saliency, attention, and graph-type explainers (respectively addressed in Sections 2, 3, and 4). These cases, developed at different timescales, encountered similar roadblocks, emphasizing shared challenges in XAI development and proving a pervasive absence of comprehensive evaluation frameworks, standardized metrics, and robust reliability guarantees in the explainable AI domain.

Even though we did not cover all XAI areas, similar stories to the ones presented exist, and the same challenges apply. For example, not-covered techniques, such as SHAP (Lundberg & Lee, 2017) and LIME (Ribeiro et al., 2016), have been demonstrated to exhibit instability and inconsis-

tency. This means that minor changes to the input can lead to significant differences in the explanations provided, and even reapplying the same method to an identical input can yield dissimilar results (Alvarez-Melis & Jaakkola, 2018; Slack et al., 2020; Lee et al., 2019). These methods also lack robust metrics for evaluating the "quality" of explanations (Lakkaraju et al., 2022).

### 5.2. Missing method-task link

Despite the abovementioned limitations, AI explainability methods continue to be applied in various contexts, propelled by the increasing demand for transparency, accountability, and trust in AI systems.

Effectively pairing XAI methods with problems they can address presents its own unique challenge. Currently, there is a scarcity of guidelines indicating which method or combination of methods would be best suited for a specific problem or dataset (Han et al., 2022). While numerous XAI surveys and taxonomies of developed methods exist, they primarily focus on technical categories (e.g., gradient vs. attention methods) rather than on categories that would suggest dataset and problem-specific XAI methods (e.g., debugging, decision making, spurious correlations), assuming they exist.

As an example, in a recent survey of XAI for healthcare (Jin et al., 2022) also used technical categories and stated: *"At the current state of the art, which method we should choose still does not have a definite answer."*

Furthermore, choosing any explanation method is not a viable option, as existing methods often disagree on the explanations they provide (Neely et al., 2021; Krishna et al., 2022).

## 6. Towards provably useful XAI

In this section, we analyze what conceptual and methodological breakthroughs are needed in order to break the recurring pattern blocking XAI methods development.

We hypothesize that one of the main challenges between the current state of XAI and provably useful XAI methods is the missing method-task link.

To be able to guarantee the method's usability and reliability in a specific task, we need either to root explanations deeply in theory directly corresponding to the task's requirements or to create use-case-inspired explanations and then empirically evaluate them in the targeted application.

### 6.1. Task-relevant theoretical guarantees

While many current methods are inspired by some mathematical concepts, these concepts are not directly relevant to

a task. For example, if a method assigning importance to individual features has a theoretical guarantee that, in the case of a linear model, it recovers the true model's coefficients, that might be a desirable mathematical behavior, but it might be utterly irrelevant for an actual task involving a complex model.

Two examples of frequently applied methods with strong mathematical underpinnings are Integrated Gradients (Sundararajan et al., 2017) and SHAP (Lundberg & Lee, 2017). Recent research (Bilodeau et al., 2022) indicates, however, that the mathematical "guarantees" result in these methods provably failing at practical applications. Specifically, (Bilodeau et al., 2022) show that *"for any feature attribution method that satisfies the completeness and linearity axioms, users cannot generally do better than random guessing for end-tasks such as algorithmic recourse and spurious feature identification."*

As pointed out by (Bilodeau et al., 2022), such methods are also considered across healthcare applications, for example, for ICU false alarm reduction task (Zaeri-Amirani et al., 2018) or cancer prognosis and diagnosis (Zhou et al., 2021; Roder et al., 2021). Therefore it is of great practical significance that users understand that common conceptions related to outputs of these methods may not hold, *"for instance, positive feature attribution does **not**, in general, imply that increasing the feature will increase the model output. Similarly, zero feature attribution does **not**, in general, imply that the model output is insensitive to changes in the feature."*

If a theory is to guarantee the method's reliability, we need to ensure that this theory is rooted in a specific task, holds in circumstances under which the method is practically applied, and that its assumptions and limitations are clear to the user.

### 6.2. Task-inspired development and evaluation

Another way to ensure the method's usefulness and reliability is through a rigorous evaluation on a real-world task of interest.

There exist some evaluation procedures for current methods; however, existing evaluation is really limited. XAI evaluations involving humans often rely on simplistic proxy tasks or subjective opinions on explanations quality (Ribeiro et al., 2016; Lundberg & Lee, 2017; Ribeiro et al., 2018; Jeyakumar et al., 2020). A recent study reveals a positive impact of explanations, specifically that they can *"reduce overreliance on AI systems during decision-making"* (Vasconcelos et al., 2023). In this study, the decision-making task involved finding a reachable exit in a maze, with a model suggesting a correct exit either with or without an explanation that described or visualized a suggested path (correct or incorrect). However, the generalizability of this

conclusion remains uncertain, as both the problem setup and the explanations were simplistic and synthetically generated. Another issue is the lack of distinction in some studies between the performance of a model with and without explanations. For instance, the highly cited work *"Explainable Machine-Learning Predictions for the Prevention of Hypoxaemia during Surgery"* (Lundberg et al., 2018) demonstrates in a study involving five anesthesiologists that doctors' predictions improve when provided with model suggestions and corresponding explanations. However, the evaluation does not include the baseline of using only the model without explanations.

Human expert studies on real-world tasks represent the gold standard for evaluations (Doshi-Velez & Kim, 2017); however, despite efforts to reduce their cost for XAI tests, they remain resource-intensive (Chen et al., 2022). Even when conducted, obtaining definitive answers can be challenging. For example, studies by (Jesus et al., 2021; Amarasinghe et al., 2022) attempt to address the lack of application-grounded XAI evaluations by assessing a few explanation methods for credit card fraud detection. Both studies focus on the same problem of identifying fraudulent transactions, and both recruit the same group of experts. However, a slight change in the experimental setup leads to contrasting conclusions: (Jesus et al., 2021) report better metrics for a model with explanations compared to one without, whereas (Amarasinghe et al., 2022) do not observe a significant difference.

In summary, it is challenging and expensive to compare and evaluate non-task-specific methods, and it is unfeasible to estimate methods' reliability for all possible real-world tasks.

However, if a method were initially developed with a specific real-world task application target and needed to be evaluated only for this application, then rigorous and empirical evaluation under realistic settings would be possible and lead to significant conclusions.

## 7. Conclusions

Given the present state of XAI, explanations without solid evidence for enhancing human-AI collaboration should not be used to justify decisions or establish trust in machine learning models. To transform the dynamics of explainability method development, a stronger focus on practical applications is crucial to avoid following the presented XAI development pattern.

We suggest treating explanations without explicit task-relevant guarantees as black boxes themselves. They can be useful but should not be trusted.

## 8. Acknowledgements

We thank Jure Leskovec and Carlos Guestrin for the discussions that inspired the work on this paper. AC was supported by Stanford School of Engineering Fellowship.

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
