# OpenReview forum: "Is Task-Agnostic Explainable AI a Myth?"
_ICML.cc/2023/Workshop/IMLH — IMLH 2023 Oral_

### Official Review · Reviewer_CBHb · 2023-06-11

**Rating:** 7
**Confidence:** 3

**Review:**

Summary:
This paper is a meta-analysis of problems in the field of Explainable AI (XAI), specifically explanations of input importance. They identify a problematic pattern and the difficulty of evaluating which explanation method is the best for a given task is present in 3 different settings, saliency maps for images, part of text importance and GNN explanations.

Strengths:
- Good clear writing identifying an important pattern
- Good summary of existing work and conflicting findings
- Addresses an important problem and justifiably highlights the need for end-task specific evaluation

Weaknesses:
- Proposed solutions are not very concrete
- Unclear to me how unique these findings are compared to existing survey papers etc.

---

### Official Review · Reviewer_cukK · 2023-06-18
**Interesting perspective article**

**Rating:** 7
**Confidence:** 4

**Review:**

This paper presents a framework for understanding the challenges in contemporary explainable AI (XAI) and highlights the conceptual and technical limitations of XAI methods, which can inadvertently turn them into black boxes. The authors examine three research avenues in XAI, covering image, textual, and graph data, and identify persistent roadblocks that emphasize the need for a conceptual breakthrough to ensure compatibility between XAI methods and application tasks. The paper concludes that, in the current state of XAI, explanations without solid evidence should not be used to justify decisions or establish trust in machine learning models, and suggests focusing on practical applications and treating explanations without task-relevant guarantees as black boxes. It introduces a unified view of XAI challenges in form of a recurrent three-stage pattern from method ideation to inconsistencies, emphasizing the need for further development in the field, and advocating for responsible use of XAI methods based on empirical evidence.

While still a survey or reflection paper of sorts, multiple of which already exist in XAI literature, this paper offers a unique perspective on the challenges of explainable AI (XAI). This paper stands out by providing a clear and actionable framework for identifying the issues, presenting novel insights and delineating potential directions for solutions. The article's contribution is valuable for the broader XAI community and can aid in advancing the practical utility of XAI methods.

---

### Meta-Review · Area_Chair_CmNP · 2023-06-19

**Recommendation:** Accept (Oral)
**Confidence:** 5

**Metareview:**

This position paper provides a novel framework addressing the ignored problem of task-specific XAI and its challenges. The perspective is unique and thought-provoking for the XAI community, and is insightful for the XAI applications in healthcare.

---

### Decision · Program_Chairs · 2023-06-20

Accept (Oral)